# Genome-Wide Gene Expression Analysis Reveals Unique Genes Signatures of Epithelial Reorganization in Primary Airway Epithelium Induced by Type-I, -II and -III Interferons

**DOI:** 10.3390/bios12110929

**Published:** 2022-10-26

**Authors:** Anna Erb, Ulrich M. Zissler, Madlen Oelsner, Adam M. Chaker, Carsten B. Schmidt-Weber, Constanze A. Jakwerth

**Affiliations:** 1Center of Allergy & Environment (ZAUM), Technical University of Munich and Helmholtz Center Munich, German Research Center for Environmental Health, Member of the German Center for Lung Research (DZL), CPC-M, Member of the Helmholtz I&I Initiative, 85746 Munich, Germany; 2Department of Otorhinolaryngology and Head and Neck Surgery, Medical School, Technical University of Munich, 81675 Munich, Germany

**Keywords:** microarray, gene expression analysis, airway epithelial, type-I, -II and -III Interferons, epithelial integrity, remodeling

## Abstract

Biosensors such as toll-like receptors (TLR) induce the expression of interferons (IFNs) after viral infection that are critical to the first step in cell-intrinsic host defense mechanisms. Their differential influence on epithelial integrity genes, however, remains elusive. A genome-wide gene expression biosensor chip for gene expression sensing was used to examine the effects of type-I, -II, and -III IFN stimulation on the epithelial expression profiles of primary organotypic 3D air-liquid interface airway cultures. All types of IFNs induced similar interferon-stimulated genes (ISGs): OAS1, OAS2, and IFIT2. However, they differentially induced transcription factors, epithelial modulators, and pro-inflammatory genes. Type-I IFN-induced genes were associated with cell–cell adhesion and tight junctions, while type-III IFNs promoted genes important for transepithelial transport. In contrast, type-II IFN stimulated proliferation-triggering genes associated and enhanced pro-inflammatory mediator secretion. In conclusion, with our microarray system, we provide evidence that the three IFN types exceed their antiviral ISG-response by inducing distinct remodeling processes, thereby likely strengthening the epithelial airway barrier by enhancing cross-cell-integrity (I), transepithelial transport (III) and finally reconstruction through proliferation (II).

## 1. Introduction

Important immune factors of the first line of cell-intrinsic host defense are the interferons (IFNs), which comprise type-I IFN (IFNα, IFNβ), type-II IFN (IFNγ), and type-III IFN (IFNλ1, IFNλ2, IFNλ3, IFNλ4). Biosensors such as toll-like receptors (TLRs) belong to the pathogen-recognition receptor family. TLRs sense so called pathogen-associated molecular patterns (PAMP) including viral nucleic acids and RNA and lead to IFN expression [1]. IFNs induce the transcription of IFN-stimulated genes (ISGs) such as the OAS enzymes that can degrade viral dsRNA and therefore limit viral replication [2] or protein assembly [3] or enhance the breakdown of the virus in infected or bystander cells [4].

Type-I IFNs are expressed across different cell types like plasmacytoid dendritic cells and mononuclear phagocytes, whereas dendritic cells are the major source for IFNλs [5,6]. Due to the expression of type-I IFN receptor (IFNAR) on a variety of immune and epithelial cells, type-I IFN can induce a widespread cell-intrinsic and immune cell-mediated antiviral response. Virus-infected cells release IFNs, which bind to their receptors on adjacent cells and activate the downstream JAK/STAT pathway, leading to ISG transcription. IFNλs bind to their IFNλ receptor on epithelial cells, leading to JAK/STAT pathway activation and mostly organizing the antiviral defense at the site of infection rather than driving pro-inflammatory systemic antiviral response [7]. IFNλ signaling thereby protects mucosal barriers from exaggerated inflammation by regulating the tissue damaging effects of neutrophils in intestinal organs [8]. Unlike type-I and -III IFNs, IFNγ is mainly expressed by natural killer cells and Th1 cells [9], enhancing the microbicidal function of macrophages, isotype switching in B cells and Th1 differentiation, while the effect on epithelial cells is still under investigation [10]. In addition, it is subsequently secreted by CD4^+^ T helper type-1 and by CD8^+^ cytotoxic T lymphocytes in an antigen-dependent [11] and -independent way [12], thereby fueling the adaptive immune response to the pathogen.

Each part of the respiratory tract shows a diverse and specific epithelial subtype that fulfills a different function; however, the impact of IFNs on these specializations is largely unclear. The lower airways mainly contain ciliated cells, which transport the mucus produced by goblet cells out of the airways, and columnar cells, which are important for the barrier function [13] (Figure 1). Previous studies showed that IFNγ might be able to decrease mucus production in the airway epithelium [14]. Although the cell composition varies in the airways, epithelial tight junctions provide the barrier to the airborne environment. Together with the adherens junctions, they form a dynamic structure called the apical junctional complex (AJC) [15]. When exposed to airborne environmental factors like allergens and pathogens, the epithelial barrier produces cytokines such as IL-33 or IL-25 that enhance the host’s immune defense mechanisms [16], leading to antigen-presenting cells (APCs), T cells and B cells recruitment [17]. Hence, the airway epithelial barrier is important for the host cell intrinsic defense and initiates the orchestration of the antiviral immune response.

Additionally, the composition of the inflammatory environment determines the differentiation of the epithelium. Type-I and type-II immune responses can lead to type-I-primed (E1) or type-II-primed (E2) polarization of the airway epithelium, thereby enhancing, for example, asthmatic remodeling processes [10]. The severity of diseases caused by respiratory viral infections is connected to the remodeling of the airway epithelium induced by viral and immune-mediated tissue-damaging processes. This epithelial destruction can lead to the aggravation of pneumonia, facilitate bacterial superinfections, and even culminate in acute respiratory distress syndrome (ARDS) [18]. Aside from the early effects of viral defense mechanisms, an association between IFNs and lung damage, morbidity and chronic diseases has been described [19]. In addition, COVID-19 disease severity and SARS-CoV-2 replication was associated with high levels of cytokines such as IFNγ, TNFα and IL-2, IL-4 and IL-6 [20]. The epithelial barrier hypothesis emphasizes this relevance of epithelial integrity [21]. Due to the exposure of environmental triggers, epithelial inflammation and tissue damage increases and thereby lead to pathogen susceptibility.

Using a genome-wide gene expression biosensor chip sensing the gene expression in organotypic 3D air-liquid interface cultures, we were able to investigate the influence of IFNs on the airway epithelial barrier. We investigate the similarities of the IFNs in relation to expressed ISGs in this airway epithelial model. In this study, we also highlight the differentially induced transcription factors, epithelial mediators and inflammatory genes unique to each IFN family member and highlight the importance of IFN release for epithelial integrity. Type-I and type-III IFNs stimulation positively correlate with the expression of cell–cell adhesion, tight junctions and intercellular signaling, whereas IFNγ promotes a pro-inflammatory environment and cell proliferation.

## 2. Materials and Methods

### 2.1. Air-Liquid Interface Organoid Cultures

Low–passage primary human bronchial epithelial cells (NHBE, Lonza) from six genet-ically independent donors (*n* = 6) were grown in PneumaCult-Ex Plus expansion medium (Stemcell) on Corning transwell polyester membrane cell culture inserts (precoated with 1% collagen, Merck, Kenilworth, NJ, USA) according to the manufacturer’s instruction. Medium was applied to the basal and apical chamber until cells were grown 100% confluence. An airlift was performed by removing the apical medium and the basal medium was exchanged to Pneumacult-ALI maintenance medium (Stemcell). When the transepithelial electrical resistance (TEER), measured using EVOM2 instrument (World Precision Instruments), reached the threshold of 700 Ω, cells each donor were stimulated with IFNα (300 IU/mL, Roche, Basel, Switzerland), IFNβ (100 IU/mL, Peprotech, Rocky Hill, NJ, USA), IFNγ (200 IU/mL, Promocell, Heidelberg, Germany), IFNλ1 (100 ng/mL, Biotechne, Minneapolis, MN, USA), IFNλ3 (100 ng/mL, Biotechne) or medium as control for 24 h. Each negative control was genetically matched. TEER measurement was used to identify epithelial integrity prior stimulation and samples showing a TEER >700 Ω were classified as integer. Cells were harvested and RNA was extracted using AllPrep DNA/RNA Micro Kit (Qiagen, Hilden, Germany). Figure 1 and Appendix A was created with BioRender.com (accessed on 26 August 2022).

### 2.2. Statistical Analysis

Cyanine-3 (Cy3) labeled cRNA was prepared from 25 ng RNA using the One-Color Agilent Low Input Quick Amp Labeling Kit (Agilent Technologies) accord-ing to the manufacturer’s protocol, followed by RNAeasy column purification (QIAGEN) (Appendix A, Appendix A). Dye incorporation and cRNA yield were checked with the NanoDrop ND-1000 Spectrophotometer. 0.6 µg of Cy3-labelled cRNA (specific activity >6.0 pmol Cy3/µg cRNA) was further processed and hybridized to Agilent Whole Human Ge-nome Oligo Microarrays (G4112A) for 17 h at 65 °C in a rotating Agilent hybridization oven. Slides were scanned on the Agilent DNA Microarray Scanner (G2505B) using one color scan setting for 8 × 60 k array slides (Scan Area 61 × 21.6 mm, Scan reso-lution 3µm, Dye channel is set to Green and 20 bit Tiff. Isolated RNA was hybridized and measured using SurePrint G3 Human gene exp v3 Array Kit (Agilent Technolo-gies). GeneSpring Software GX 14.9 (Agilent technologies, Santa Clara, CA, USA) was used to identify signif-icant changes in RNA expression. Genes showing a FC value ≥ 1.5 (fold change) and *p* < 0.05 by using moderated ttest were considered as significantly differentially ex-pressed hits. Due to low expression amplitude and variability of gene expression in primary cells, we did not apply multiple testing correction to the gene expression analysis. These significantly regulated genes were summarized in entity lists and GeneOntology (GO) terms “0008009”, “0005125”, “0005615” and “0007267” were used for segregation of secreted genes, “0070254”, “0097072”, “0001730”, “0035394” and the gene symbols IFITM and IFIT for interferon stimulated genes, “0009615”, “0039528”, “0039530”, “0039639”, “0051607” and “0009597” for antiviral pathways. For identification of transcription factors GO terms “0003676”, “0044212”, “0000976”, “0070491” and the gene name forkhead was applied, GO terms for identification for pro- and anti-inflammatory genes “0033209”, “0032640”, “0034612”, “0006954”, “0050727”, “0002526”, “0032611”, “0070555”, “2000661”, “0032635”, “0004915”, “0070741”, “0070103” and “0036363”, “0071604”, “0004920”, “0032613”, “0050728”, “0140105”, “0032733” was used, GO terms “0060429”, “0060428”, “0030054”, “0016049”, “0098609”, “0090136”, “0034103”, “0070160”, “0030057”, “0070160”, “0120193”, “0005923”, “0005915” and “0005921” were applied to identify epithelial factors. A 1.5-fold change above the “Normalized intensity values” in relation to the medium control were classified as high (red) abundance.

### 2.3. String Network Analysis

In order to extract enriched cellular processes and pathways associated with epithelial mediators [22] and inflammatory genes, an open-access tool, the string network analysis version 11.5 was used (string-db.org (accessed on 12 September 2022), [23,24]). The analysis was performed as described before [25]. The STRING database uses open-source information about protein-protein interactions and connect this information with computational predictions to reveal physical and functional protein interactions.

## 3. Results

### 3.1. Differential IFN-Type-Specific Secreted Factors despite Overlapping Pan-IFN-Triggered ISG Expression

IFN-stimulated organotypic 3D air–liquid interface (ALI) cultures were used to identify IFN-induced specific differentially expressed genes (DEGs). The IFNs were titrated prior to transcriptome analysis for biological activity based on a comparable expression (data not shown). All IFNs induced the upregulation of DEGs rather than their suppression (Figure 2A; fold change (FC) ≥ 1.5; *p* < 0.05). IFNα had the strongest and IFNβ stimulation the least influence on the differential gene expression (Figure 2A). Using GeneOntology (GO) annotation displaying ISGs, we confirmed and validated our ALI cultures and observed a high overlap in the induction of known ISGs by the different IFN family members (Figure 2B). All IFNs enhanced the induction of OAS1, OAS2, OAS3, OASL, IFIT2, and IFITM-1, -2 and -3 (Figure 2C), but induction by IFNβ was marginal. It is important to note that only IFNα induced the gene IFIT1B (Figure 2B). The GO-term analysis of all secreted factors revealed that four DEGs were induced by all types of IFNs: the antiviral mediators CXCL10, OAS3, SLPI and IL-36 receptor antagonist IL36RN (Figure 2D,E). More importantly, the analysis demonstrates that IFNs display distinct signatures of secreted factors (Figure 2D).

### 3.2. IFN-Type-Specific Induction of Distinct Transcription Factor Patterns in Airway Epithelia

Since type-I, -II, and –III IFNs upregulated similar ISGs and distinct secreted factors, we also analyzed transcription factors and other DNA binding elements to identify gene regulation patterns. Unexpectedly, all IFN family members induced a very diverse transcription factor pattern (Figure 3A–D). IFNα and IFNγ induced the highest variety of transcription factors (IFNα: 90 entities; IFNγ: 88 entities) compared with unstimulated cultures. Remarkably, all IFNs induced TLE1 compared with unstimulated cultures, with IFNγ showing the strongest effect on its gene expression (Figure 3B–D). In addition, IFNs of type-I and type-III induced the expression of TLE4. The expression of RNASE11, GATA4, ZNF541 and EHD4 were specifically enhanced by IFNα (Figure 2B), while IFNβ stimulation elicited the upregulation of IKZF1 and NR1H2. Lung development and cell cycle progression is regulated by genes such as FoxP2, ZBTB20, RNASE12, SOX18, and NANOG, which were induced by type-III IFNλ1 and -3 (Figure 3C). In addition, gene expression analysis showed that type-II IFN enhanced both RNASE11 and CIITA expression (Figure 3D). Taken together, these data demonstrate that each member of the IFN family induces a unique regulation pattern of transcription factors.

### 3.3. Induction of Pro-Inflammatory Rather than Homeostatic Genes by All IFNs Except IFNλ3

To determine whether the release of IFN leads to a pro-inflammatory or homeostatic constitution of the airway epithelium, cultures were examined for pro- and anti-inflammatory factors following the GO annotation. In fact, IFN stimulation enhanced pro- rather than homeostatic or anti-inflammatory genes (Figure 4A,B). In comparison, type-II and -III IFNs induced 23 equal pro-inflammatory DEGs, while type-I had eight overlapping DEGs with type-II and -III IFNs. In addition, all IFN differentially regulated anti-inflammatory genes (Figure 4B). This analysis also confirmed that only type-I IFNs and IFNγ regulated METRNL, while type-III IFNs showed no overlapping induced anti-inflammatory genes with the other IFNs. On the one hand, the induction of IL36RN, STAT1, USP18 and CXCL10 was observed in all type-I, -II, and –III IFN-stimulated ALI cultures compared to unstimulated cultures (Figure 4C–E). On the other hand, IFNλ1 had less effect on the regulation of these pro-inflammatory DEGs (CXCL10 and IL36RN) compared with unstimulated cultures and the other IFN-stimulated cultures. In addition, IFNα and IFNγ enhanced the expression of pro-inflammatory genes such as CXCL9, CXCL11, GBP5, CD38, JAK2 and APOL3 compared with unstimulated cultures (Figure 4C,E). In contrast, IFNβ induced pro-inflammatory DEGs such as AOC3, H2BC1 and CD70, while IFNλ3 upregulated AKNA and the chemokine CXCL13. Notably, IFNγ downregulated pro-inflammatory genes such as TSLP and PBK compared to unstimulated cultures, while genes associated with cell proliferation and differentiation such as ACP5, CALCRL, and PTPN2 were upregulated (Appendix A). All analyzed IFNs also upregulated mildly anti-inflammatory genes like IL12B and IL22RA2, with IFNλ3 showing the strongest impact on anti-inflammatory DEG expression by uniquely upregulating the regulatory cytokine IL10 (Appendix A). Type-I, -II and -III IFN stimulation resulted in differential chemokine expression, and IFNα and IFNγ in particular enhanced the expression of pro-inflammatory genes. Simultaneously, IFN release, but especially IFNλ3 stimulation, led to an induction of homeostatic, anti-inflammatory factors.

### 3.4. Enhancement of Cell-Cell Adhesion by Type-I, of Intercellular Signaling and Ion Channels by Type-III and of Cell Growth by Type-II IFNs

To compare the effects of type-I, -II, and -III IFNs on airway epithelial structure after a viral infection, GO terms were used to identify structural epithelial genes. All IFN family members upregulated genes associated with the tight junctions and cell-cell adhesion (PATJ and JAM). Type-I and type-II IFNs showed matching effects only in upregulating CBLL1, while type-I and type-III IFNs similarly induced PECAM1 (Figure 5A). Furthermore, IFNγ was consistent with type-III IFNs in the induction of seven similarly regulated epithelial DEGs (Figure 5A). In particular, type-I IFNs upregulated genes important for epithelial integrity such as cell–cell adhesion and tight junctions compared to unstimulated cultures: ICAM2, PECAM1, CDH10, TJP2, JAML, CLDN25 and CBLL1 (Figure 5B). In addition, type-III IFNs induced genes associated with intercellular signaling via ion channels and receptors such as CTNNA2, GABRR1, GABRA2, GLRA3, SYT9, and GABRG3 (Figure 5C). Specifically, IFNγ upregulated DEGs associated with cell growth, the complement system, and inflammation: DUOX2, PTPRR, C4A, C4B and HTR20 (Figure 5D). Conclusively, these data showed the differential impact on epithelial integrity of the three IFN types in relation to the regulation of cell–cell adhesion, tight junctions, ion channels and cell growth.

### 3.5. IFN-Induced Hub DEGs Connect Signaling Pathways in the Airway Epithelium

To identify enriched cellular gene networks induced by type-I, -II, and –III IFNs, we performed a network analysis using the “STRING” database [24], In particular, the analysis included genes associated with inflammatory pathways as well as epithelial integrity and cell proliferation. Although CXCL10, STAT1, PSMB8 and USP18 (Figure 6A, grey) were induced by all IFNs, pathway analysis revealed uniquely regulated genes, specifically by each member of the IFN family (Figure 6). Type-I IFN (red) induced TGFB1 and its regulator Latent TGF Beta Binding Protein 3 LTBP3 as well as IL-6 and IL-18 receptors IL18R1 and IL18RAP (Figure 6A). Chemotactic factors such as CCL7, CCL8, CCL19 and CCL25 as wells as IL1b were induced by type-II IFN (purple). The last cluster (yellow) consisted of interleukins such as IL-9, IL-10 and the receptors TNFα receptor TNFRSF1B and IL-6 receptor IL6R, all of which were only induced by type-III IFN. It is important to note that no IFN-induced overlapping gene regulation of genes associated with epithelial remodeling or integrity with any other IFN. Gene network analysis revealed three clearly distinguishable groups: cluster 1 (red) mainly comprised cell-cell adhesion and tight junction-associated genes induced by type-I IFNs such as JAM2, VCL and PECAM1, cluster 2 (purple) contained various growth factors and genes associated with cytoskeletal organization (KDR, ARHGEF5) mainly induced by type II IFN (Figure 6B), while cluster 3 specifically connected tight junctions and cell-cell adhesion molecules as well as intercellular signaling molecules induced by type-III IFNs (TJP1, IGSF5, CTNNA1-3, NRXN1) (Figure 6B, yellow).

In summary, all IFNs induced different gene expression patterns in the airway epithelium and enhanced several important factors related to inflammatory pathways and epithelial remodeling. Factors associated with beneficial effects on epithelial barrier integrity by enhancing cell–cell adhesion, tight junctions and intercellular signaling were induced by type-I and -III IFNs, while type-II IFN stimulation resulted in cell growth and a pro-inflammatory environment.

## 4. Discussion

Although the mechanisms of the biosensors upstream of the interferon induction and the role of interferons in antiviral signaling pathways are well understood, their effect on airway epithelial processes is widely unknown. In this study, we provide evidence for the unique effect of each member of the IFN family on epithelial homeostasis, remodeling, and the inflammatory response of the human airway using genome-wide gene expression analysis of IFN-stimulated primary organotypic air-liquid interface cultures. In these cultures, all IFNs induced ISGs, as expected, but notably, the number of ISGs induced between individual IFN family members widely overlapped. The induced ISGs interfere with the viral entry (IFITM), viral mRNA synthesis (APOBECs), protein synthesis (IFIT1) and viral replication (APOBECs, OAS) [4], In this study differential expression of transcription factors, pro-inflammatory molecules, and epithelial mediators were analyzed which are unique to each member of the IFN family were analyzed. It became apparent that the three types of IFN enhanced the proinflammatory environment and stimulated three distinct epithelial mechanisms: type-I IFNs induce cell-cell adhesion molecules and tight junctions, type-III IFN induced ion channels and intercellular processes and type-II IFN-stimulated cells expressed genes associated with cell proliferation. In addition, the IFN family members stimulated distinct the proinflammatory environment.

In particular, type-I IFN stimulation, specifically IFNα, induced the upregulation of the pro-inflammatory gene *IL18R1*, whose expression was associated with asthma severity in human bronchial epithelial biopsy and in bronchial alveolar lavage [26]. In addition, pro-inflammatory ISGs were reported in the respiratory tract of COVID-19 patients [27]. It is important to note that only IFNα-stimulated ALI cultures induced upregulation of *GATA4*, which has been shown to play a role in early development of the lung mesenchyme [28]. In addition, type-I IFNs showed an impact on the epithelial airway structure by differentially induced expression of cell-cell adhesion molecules and tight junctions (*ICAM2*, *PECAM1*, *JAM2*, *TJP2*). Previous studies have shown that IFN-enhanced epithelial resistance to influenza infection [29], established a resistant blood–brain barrier in in vivo models [30], and supported a barrier to bacterial dissemination [31]. Patients suffering from house dust mite-induced allergic rhinitis showed an increased epithelial permeability and decreased occludin expression [32]. Our study showed that especially type-I IFN induced tight junctions (*CLDN9*, *CLDN25*, *JAML*, *PATJ*). In addition, various studies have highlighted that *JAM2* and *PECAM1* are downregulated by the SARS-CoV-2 virus, which resulted in disruption of the alveolar–capillary barrier [33]. This study demonstrates that type-I IFNs induce specific transcription factors and genes associated with cell-cell adhesion and tight junctions. We propose that the early release of type-I IFNs support the host’s intrinsic virus defense mechanisms by strengthening the epithelial barrier and preventing transcendence of virus particles.

Despite the postulated similar mode of action of type-I and -III IFNs in antiviral host defense mechanisms, type-III IFNs stimulated a unique epithelial gene expression and resulted in a secondary process: the induction of anti-inflammatory gene expression and intercellular signaling. IFNλ3 induced anti-inflammatory factors such as *IL10* in differentiated airway epithelium. Previous studies have shown that IL-10 activation in dendritic cells and macrophages is dependent on type-I IFN signaling [34]. It is interesting to note that here, however, only IFNλ3 stimulates the expression of IL-10 in the differentiated airway epithelial cells. In addition, type-III IFNs induced transcription factors such as *FoxP2* and *ZBTB20*. Previous studies showed that processes during lung development and cell cycle progression regulated by Cyclin D1, Cyclin E and P21 are associated with FoxP2 and ZBTB20 [35]. In addition, Type-III IFNs stimulated the expression of several ion channels, receptors and adhesion molecules (*GABRR1*, *GABRA2*, *GLRA3*, *SYT9*, *GABRG3* and *CTNNA1-3*), that may differentially regulate both intracellular signaling processes and transepithelial and cross-epithelial ion transport along the barrier [36]. In addition, these IFN family members might increase cell permeability via NF-κB activation by inducing the TNFα receptor *TNFRSF1B* [37]. These data show that type-III IFNs contribute to the regulation of transepithelial transport and signaling processes during viral infections by inducing ion channels, receptors and adhesion molecules.

Our data showed that IFNα and IFNγ in particular shared the overlapping induction of pro-inflammatory genes such as *CXCL9*, *CXCL11*, *GBP5*, *CD38*, *JAK2* and *APOL3*. This finding is in line with previously published mechanisms of type-I and type-II interferons shown for acute influenza A virus infection [38] and HSV-2 virus infection [39], where the orchestrated stimulation and release of type-I and -II IFNs resulted in APC activation and an adaptive type 1 T helper (Th1) response. Mainly type-II IFN has been identified as the main driver of the cytokine storm in COVID-19 patients [40] and is associated with cytokine dysfunction of secondary organs [41].

It is important to note that our data also provide evidence for a unique expression pattern in type-II IFN-stimulated cultures that result in a third epithelial process. IFNγ alone differentially induced genes such as *IL1B*, *CCL7*, *KDR* and *ARHGEF5* indicative of a pro-inflammatory environment and cell proliferation. IL-1β, activated by the inflammasome, may dysregulate the epithelial barrier function in neutrophilic asthma and lead to increased mucin production and neutrophilic infiltration [42]. High levels of kinase insert domain receptor (KDR) along with its ligand, vascular endothelial growth factor A (*VEGFA*), have been associated with shorter survival times in lung cancer patients due to their effects on cell proliferation [43], as has been *ARHGEF5*, which leads to cell invasion and tumor growth via Akt pathway [44]. Therefore, in addition to its stimulation of pro-inflammatory mediators, we show a unique effect of type-II IFNs on airway epithelial cell proliferation. Future studies will have to confirm the role of these epithelial mediators and inflammatory genes in the IFN dependent antiviral cell state and the involved innate immune response.

Our data suggest that type-I IFNs improve cross-cell integrity by enhancing cell–cell and gap junctions, thereby directly counteracting viral infection-induced airway barrier leakage. Type-III IFNs strengthen the epithelial barrier by enhancing transepithelial transport. Type-II IFNs, on the other hand, promote a pro-inflammatory microenvironment and stimulate epithelial cell proliferation, likely contributing to restoration of the airway barrier integrity after viral infection.

This study not only demonstrates the relevance of genome-wide gene expression analysis using differentiated organotypic 3D air-liquid interface cultures but also showed that the different members of the IFN family have unique effects on airway epithelial barrier integrity in addition to their direct antiviral ISG-mediated functions. Future personalized inhalant therapies may profit from this study and target IFN-type-specific disfunction of epithelial integrity and support specific intrinsic anti-viral immunity. In particular, immunocompromised and elderly patients may profit from this personalized therapeutic strategy. Future studies need to examine long-term effects of these important antiviral mediators on the airway epithelium also post-infection, since an altered restoration process of the airway epithelium could contribute to adverse events as observed in patients with long COVID.

## Figures and Tables

**Figure 1 biosensors-12-00929-f001:**
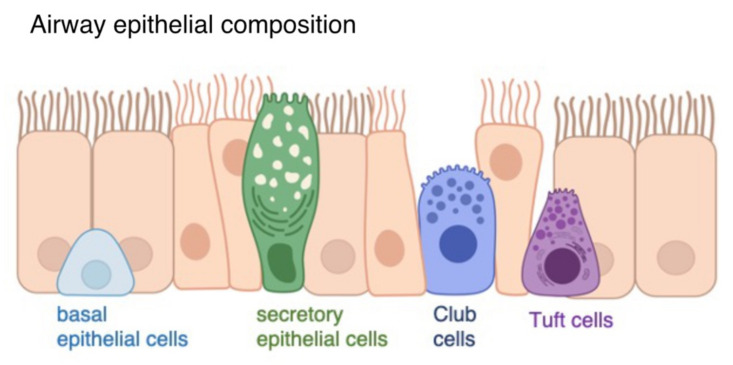
Airway epithelial composition. The airway epithelium consists of ciliated cells (orange), basal epithelial cells (light blue), secretory epithelial cells (green), Club cells (dark blue) and tuft cells (purple).

**Figure 2 biosensors-12-00929-f002:**
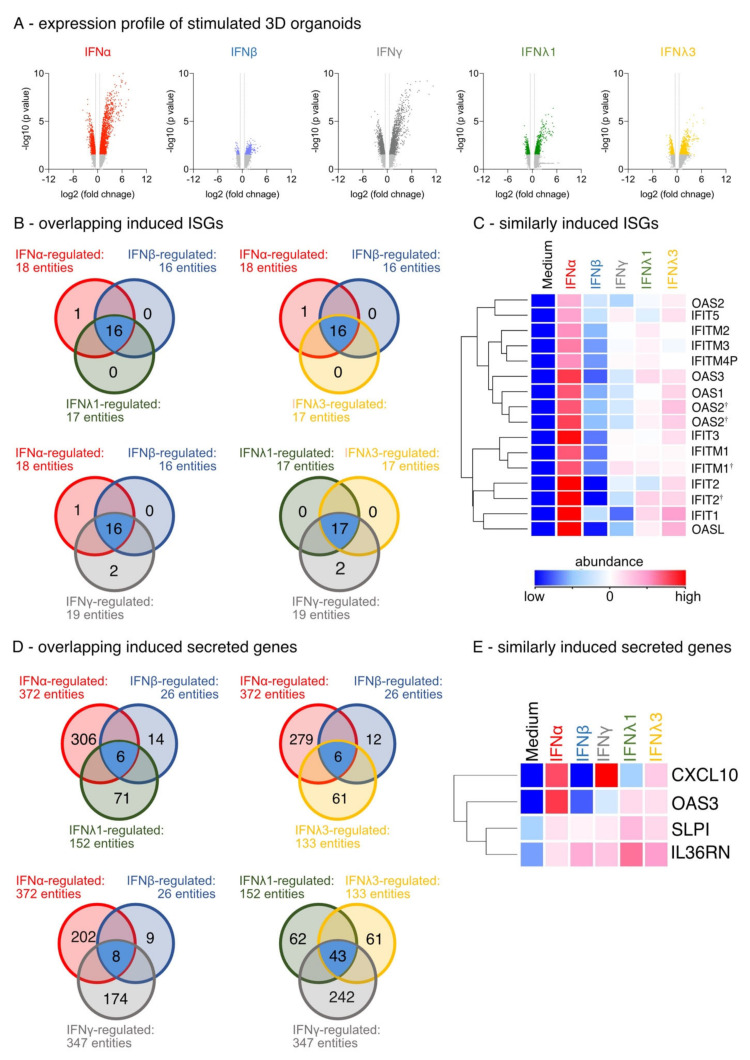
Expression profile of IFN-stimulated differentiated organotypic 3D air-liquid interface cultures. (**A**) Volcano Blot of regulated genes (fold change (FC) ≥ 1.5; *p* < 0.05). Red: ttest IFNα-stimulated vs. unstimulated, blue: ttest IFNβ-stimulated vs. unstimulated, grey: ttest IFNγ-stimulated vs. unstimulated, green: ttest IFNλ1-stimulated vs. unstimulated, yellow: IFNλ3-stimulated vs. unstimulated. (**B**) Venn blot of overlapping regulated ISGs by type-I, -II, and -III IFNs. Ttest IFNα-stimulated vs. ttest IFNβ-stimulated and ttest IFNλ1-stimulated. Ttest IFNα-stimulated vs. ttest IFNβ-stimulated and ttest IFNλ3-stimulated vs. unstimulated. Ttest IFNα-stimulated vs. ttest IFNβ-stimulated and ttest IFNγ-stimulated. Ttest IFNλ1-stimulated vs. ttest IFNλ3-stimulated and ttest IFNγ-stimulated. Blue: overlapping DEGs. (**C**) Heat map of overlapping upregulated ISGs by type-I, -II, and -III IFN. Duplicate gene names indicate the abundance of two or more transcripts of the same gene in the analysis and are marked with a cross. (**D**) Venn blot of overlapping regulated secreted factors by type-I, -II, and -III IFNs. (**E**) Heat map of overlapping upregulated secreted factors by type-I, -II, and -III IFN.

**Figure 3 biosensors-12-00929-f003:**
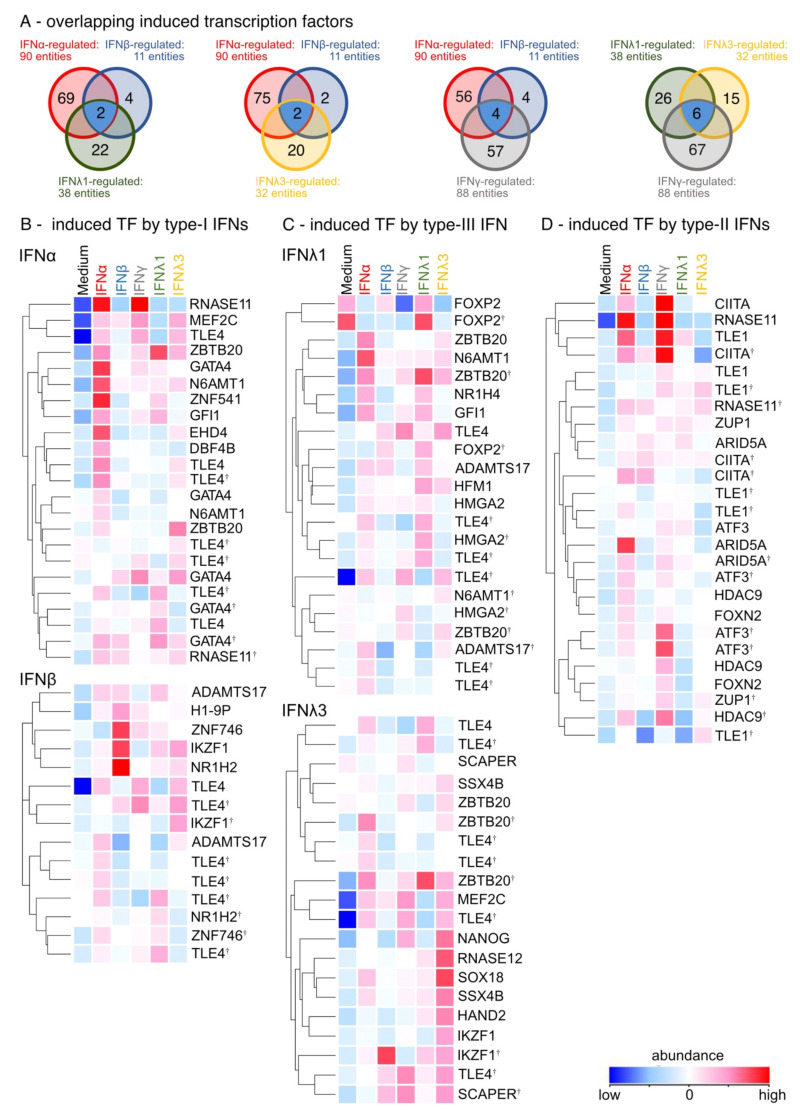
IFN family members differentially induce transcription factors. (**A**) Venn blot of overlapping regulated transcription factors by type-I, -II and -III IFNs. Ttest IFNα-stimulated vs. ttest IFNβ-stimulated and ttest IFNλ1-stimulated. Ttest IFNα-stimulated vs. ttest IFNβ-stimulated and ttest IFNλ3-stimulated vs. unstimulated. Ttest IFNα-stimulated vs. ttest IFNβ-stimulated and ttest IFNγ-stimulated. Ttest IFNλ1-stimulated vs. ttest IFNλ3-stimulated and ttest IFNγ-stimulated. Blue: overlapping DEGs. (**B**) Heat map of induced transcription factors by type-I IFNs (ttest IFNα-stimulated; ttest IFNβ-stimulated vs. unstimulated). Duplicate gene names indicate the abundance of two or more transcripts of the same gene in the analysis and are marked with a cross. (**C**) Heat map of induced transcription factors by type-III IFNs (ttest IFNλ1-stimulated; IFNλ3-stimulated vs. unstimulated). (**D**) Heat map of induced transcription factors by type-II IFNs (ttest IFNγ-stimulated vs. unstimulated).

**Figure 4 biosensors-12-00929-f004:**
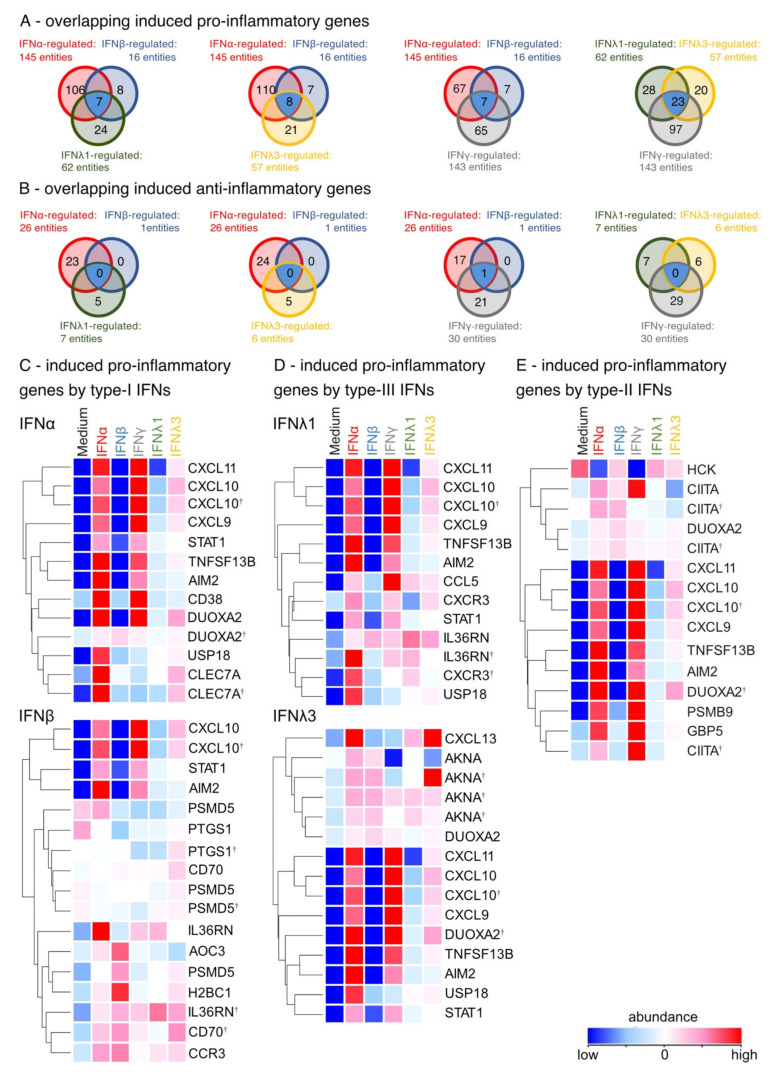
IFN stimulation leads to differentially expressed pro- and anti-inflammatory genes. (**A**) Venn blot of overlapping regulated pro-inflammatory genes by type-I, -II and -III IFNs. Test IFNα-stimulated vs. ttest IFNβ-stimulated and ttest IFNλ1-stimulated. Ttest IFNα-stimulated vs. ttest IFNβ-stimulated and ttest IFNλ3-stimulated vs. unstimulated. Ttest IFNα-stimulated vs. ttest IFNβ-stimulated and ttest IFNγ-stimulated. Ttest IFNλ1-stimulated vs. ttest IFNλ3-stimulated and ttest IFNγ-stimulated. Blue: overlapping DEGs. (**B**) Venn blot of overlapping regulated anti-inflammatory genes by type-I, -II and -III IFNs. Ttest IFNα-stimulated vs. ttest IFNβ-stimulated and ttest IFNλ1-stimulated. Ttest IFNα-stimulated vs. ttest IFNβ-stimulated and ttest IFNλ3-stimulated vs. unstimulated. Ttest IFNα-stimulated vs. ttest IFNβ-stimulated and ttest IFNγ-stimulated. Ttest IFNλ1-stimulated vs. ttest IFNλ3-stimulated and ttest IFNγ-stimulated. Blue: overlapping DEGs. (**C**) Heat map of induced pro-inflammatory genes by type-I IFNs (ttest IFNα-stimulated; ttest IFNβ-stimulated vs. unstimulated). Duplicate gene names indicate the abundance of two or more transcripts of the same gene in the analysis and are marked with a cross. (**D**) Heat map of induced pro-inflammatory genes by type-III IFNs (ttest IFNλ1-stimulated; IFNλ3-stimulated vs. unstimulated) (**E**) Heat map of induced pro-inflammatory genes by type-II IFNs (ttest IFNγ-stimulated vs. unstimulated).

**Figure 5 biosensors-12-00929-f005:**
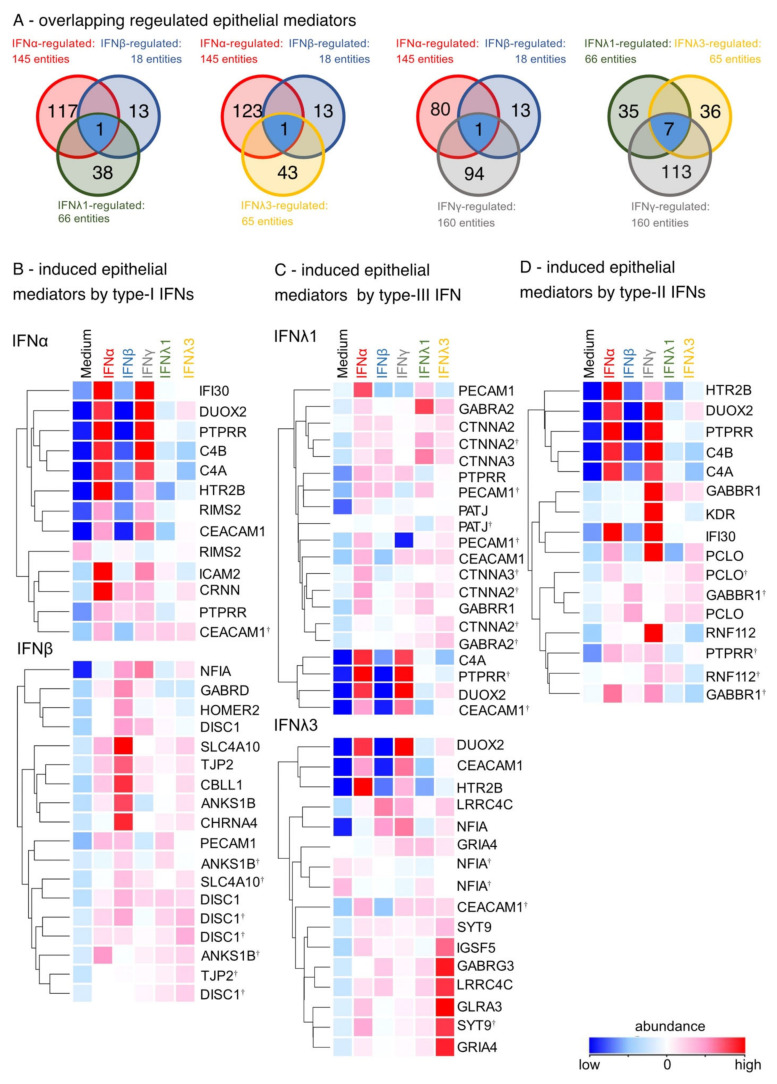
Type-I, -III and -II IFNs enhance the differentially induction of epithelial mediators. (**A**) Venn blot of overlapping regulated epithelial mediators by type-I, -II and -III IFNs. Ttest IFNα-stimulated vs. ttest IFNβ-stimulated and ttest IFNλ1-stimulated. Ttest IFNα-stimulated vs. ttest IFNβ-stimulated and ttest IFNλ3-stimulated vs. unstimulated. Ttest IFNα-stimulated vs. ttest IFNβ-stimulated and ttest IFNγ-stimulated. Ttest IFNλ1-stimulated vs. ttest IFNλ3-stimulated and ttest IFNγ-stimulated. Blue: overlapping DEGs. (**B**) Heat map of induced epithelial mediators by type-I IFNs (ttest IFNα-stimulated; ttest IFNβ-stimulated vs. unstimulated). Duplicate gene names indicate the abundance of two or more transcripts of the same gene in the analysis and are marked with a cross. (**C**) Heat map of induced epithelial mediators by type-III IFNs (ttest IFNλ1-stimulated; IFNλ3-stimulated vs. unstimulated). (**D**) Heat map of induced epithelial mediators by type-II IFNs (ttest IFNγ-stimulated vs. unstimulated).

**Figure 6 biosensors-12-00929-f006:**
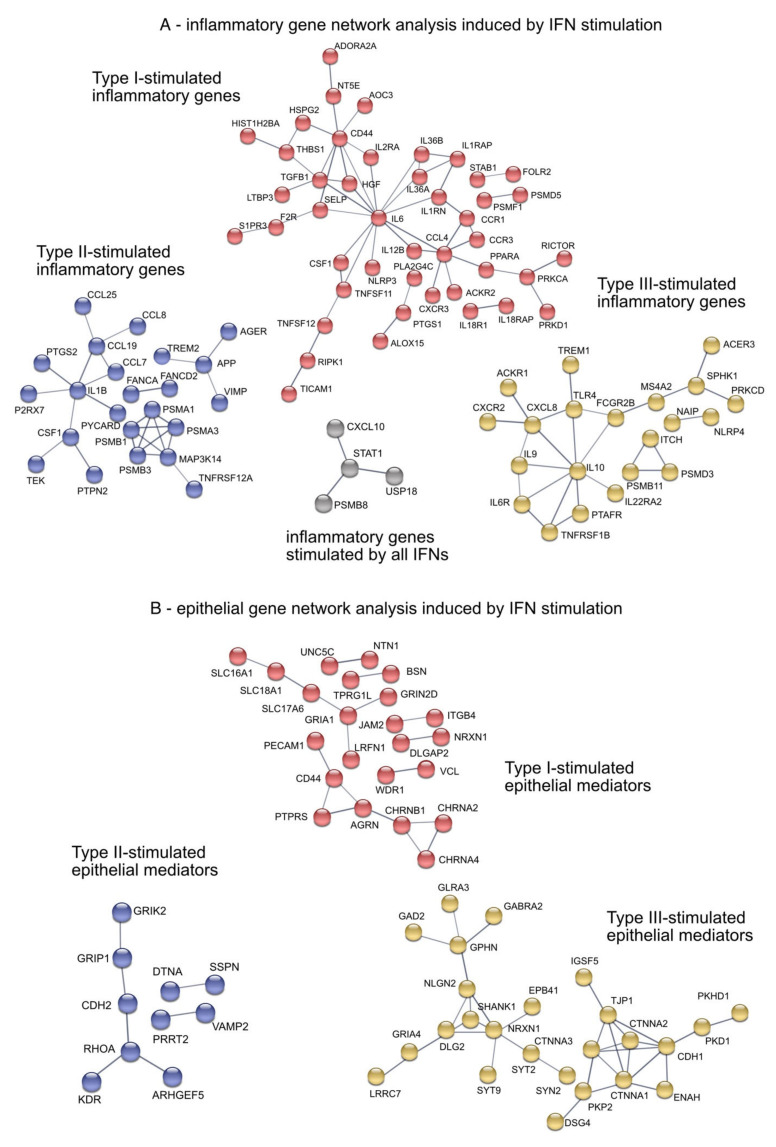
Gene network analysis of inflammatory pathways and epithelial mediators of IFN-stimulated differentiated organotypic 3D air-liquid interface cultures. (**A**) Pro-inflammatory gene network analysis. Grey: overlapping induced inflammatory genes by type-I, -II and -III IFNs. Red: Type-I IFNs-induced inflammatory genes. Purple: Type-II IFN-induced inflammatory genes. Yellow: Type-III IFNs-induced inflammatory genes (**B**) Epithelial mediators gene network analysis. Red: Type-I IFNs-induced epithelial mediators. Purple: Type-II IFN-induced epithelial mediators. Yellow: Type-III IFNs-induced epithelial mediators.

## Data Availability

The data discussed in this publication are deposited in NCBI’s Gene Expression Omnibus and are accessible under the GEO Series accession number GSE209727.

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
