# Peer review of "Genome-Wide Gene Expression Analysis Reveals Unique Genes Signatures of Epithelial Reorganization in Primary Airway Epithelium Induced by Type-I, -II and -III Interferons"

_biosensors, 2022, doi:10.3390/bios12110929_

Round 1

Reviewer 1 Report

The manuscript describes the confirmation by a microarray system that three IFN types exceed their antiviral ISG response by inducing different remodeling processes, thus potentially strengthening the epithelial airway barrier by enhancing cross-cell-integrity (I), transepithelial transport (III) and finally remodeling by proliferation (II). This work is interesting and well designed. To make this work more readable and scientific, the following sections are some of our suggestions.

The authors introduced the function of interferons (IFNs) and the importance of respiratory epithelial cells in the introduction section, which could have been made more readable if this section could have been abbreviated.

In the discussion section the authors describe three types of interferons (IFNs) that enhance the pro-inflammatory environment and stimulate three different epithelial mechanisms. A fuller explanation of this section and elucidation of its role would have made the article more readable.

The manuscript has less description of microarray, so the Schematic diagram of microarray is added and the advantages of using microarray are explained to make the manuscript more complete.

We suggest adding a description of the social significance of the research in the discussion part to improve the content of the article.

Author Response

Point-by-point Response

Date: 10/19/2022

Manuscript Number: biosensors-1945460

Title of Article: Genome-wide gene expression analysis reveals unique genes signatures of epithelial reorganization in primary airway epithelium induced by type-I, -II and -III Interferons

Name of Corresponding Author: Constanze A. Jakwerth, PhD

Email Address of Corresponding Author: constanze.jakwerth@tum.de

Reviewer #1

The manuscript describes the confirmation by a microarray system that three IFN types exceed their antiviral ISG response by inducing different remodeling processes, thus potentially strengthening the epithelial airway barrier by enhancing cross-cell-integrity (I), transepithelial transport (III) and finally remodeling by proliferation (II).

This work is interesting and well designed.

We thank the reviewer for the positive feedback. Please find our response to each question below with line numbers corresponding to the lines in the clean version.

To make this work more readable and scientific, the following sections are some of our suggestions. The authors introduced the function of interferons (IFNs) and the importance of respiratory epithelial cells in the introduction section, which could have been made more readable if this section could have been abbreviated.

We thank the reviewer for this suggestion and have adapted the Introduction section (Line 45; 49; 52; 61; 64; 66): “Subsequently, the STAT1/STAT2/IRF9 complex binds to IFN-stimulated response elements (ISRE) and activates ISG transcription.”; “Activation of type-I and type-III IFNs result in positive feedback loops, since they induce IRF7 and IRF1, which in turn initiate the transcription of IFNAR1.”; “The cytokine is released by natural killer (NK) cells during the first innate immune response to gram-negative and LPS positive bacteria.”; “Between the alveolar space and the airways, club cells protect the epithelium by producing factors like secretoglobin 1A1 (SCGB1A1).“; “Paracellular permeability is enhanced by AJC disruption, which leads to viral susceptibility and replication.“

In the discussion section the authors describe three types of interferons (IFNs) that enhance the pro-inflammatory environment and stimulate three different epithelial mechanisms. A fuller explanation of this section and elucidation of its role would have made the article more readable.

We appreciate this important comment, we realized that the wording in this section was not sufficiently concise. We amended the paragraph according to the reviewer’s suggestions (Line 331-347): “In this study differential expression of transcription factors, pro-inflammatory molecules, and epithelial mediators were analyzed which are unique to each member of the IFN family were analyzed. It became apparent that the three types of IFN enhanced the proinflammatory environment and stimulated three distinct epithelial mechanisms: type-I IFNs induce cell-cell adhesion molecules and tight junctions, type-III IFN induced ion channels and intracellular processes and type-II IFN-stimulated cells expressed genes associated with cell proliferation. In addition, the IFN family members stimulated distinct the proinflammatory environment. In this study differential expression of transcription factors, pro-inflammatory molecules, and epithelial mediators were analyzed which are unique to each member of the IFN family. It became apparent that the three types of IFN enhanced the proinflammatory environment and stimulated three distinct epithelial mechanisms. In particular, type-I IFN stimulation, specifically IFNα, induced the upregulation of the pro-inflammatory gene IL18R1, whose expression was associated with asthma severity in human bronchial epithelial biopsy and in bronchial alveolar lavage [27]. In addition, pro-inflammatory ISGs were reported in the respiratory tract of COVID-19 patients [49]. It is important to note that only IFNα-stimulated ALI cultures induced upregulation of GATA4, which has been shown to play a role in early development of the lung mesenchyme [28]. In addition, type-I IFNs showed an impact on the epithelial airway structure by differentially induced expression of cell-cell adhesion molecules and tight junctions (ICAM2, PECAM1, JAM2, TJP2)”

The manuscript has less description of microarray, so the Schematic diagram of microarray is added and the advantages of using microarray are explained to make the manuscript more complete.

We have included a more detailed description of the microarray technique in the Methods section (Line 116-145): “Cyanine-3 (Cy3) labeled cRNA was prepared from 25 ng RNA using the One-Color Agilent Low Input Quick Amp Labeling Kit (Agilent Technologies) ac-cord-ing to the manufacturer's protocol, followed by RNAeasy column purification (QIAGEN) (SupFig.1). Dye incorporation and cRNA yield were checked with the NanoDrop ND-1000 Spectrophotometer. 0.6 µg of Cy3-labelled cRNA (specific activity >6.0 pmol Cy3/µg cRNA) was further processed and hybridized to Agilent Whole Hu-man Ge-nome Oligo Microarrays (G4112A) for 17 hours at 65°C in a rotating Agilent hybridization oven.  Slides were scanned on the Agilent DNA Microarray Scanner (G2505B) using one color scan setting for 8x60k array slides (Scan Area 61x21.6 mm, Scan reso-lution 3µm, Dye channel is set to Green and 20 bit Tiff. Isolated RNA was hybridized and measured using SurePrint G3 Human gene exp v3 Array Kit (Agilent Technolo-gies). GeneSpring Software GX 14.9 (Agilent technologies) was used to iden-tify signif-icant changes in RNA expression. Genes showing a FC value ≥ 1.5 (fold change) and p < 0.05 by using moderated ttest were considered as significantly differ-entially ex-pressed hits. Due to low expression amplitude and variability of gene ex-pression in primary cells, we did not apply multiple testing correction to the gene ex-pression analysis. These significantly regulated genes were summarized in entity lists and GeneOntology (GO) terms “0008009”, “0005125”, “0005615” and “0007267” were used for segregation of secreted genes, “0070254”, “0097072”, “0001730”, “0035394” and the gene symbols IFITM and IFIT for interferon stimulated genes, “0009615”, “0039528”, “0039530”, “0039639”, “0051607” and “0009597” for antiviral pathways. For identification of transcription factors GO terms “0003676”, “0044212”, “0000976”, “0070491” and the gene name forkhead was applied, GO terms for identification for pro- and anti-inflammatory genes “0033209”, “0032640”, “0034612”, “0006954”, “0050727”, “0002526”, “0032611”, “0070555”, “2000661”, “0032635”, “0004915”, “0070741”, “0070103” and “0036363”, “0071604”, “0004920”, “0032613”, “0050728”, “0140105”, “0032733” was used, GO terms “0060429”, “0060428”, “0030054”, “0016049”, “0098609”, “0090136”, “0034103”, “0070160”, “0030057”, “0070160”, “0120193”, “0005923”, “0005915” and “0005921” were applied to identify epithelial factors. A 1.5-fold change above the “Normalized intensity values” in relation to the medium control were classified as high (red) abundance.”

In addition, we have included a schematic illustration (Suppl.Fig.1) to facilitate the understanding for the readership, as suggested by the reviewer.

We suggest adding a description of the social significance of the research in the discussion part to improve the content of the article.

We appreciate this suggestion and have now supplemented the discussion with the following sentence (Line 408-412): “Future personalized inhalant therapies may profit from this study and target IFN-type-specific disfunction of epithelial integrity and support specific intrinsic antiviral immunity. In particular, immunocompromised and elderly patients may profit from this personalized therapeutic strategy.”

Reviewer 2 Report

The authors report differential gene expression and protein-protein interaction analyses of microarray data obtained from an airway epithelial model treated with different interferon molecules. While this reviewer agrees that understanding the impact of interferons in terms of antiviral activity and epithelial reorganization is important and useful to our readership, I think the manuscript is missing a few pieces of key information. Therefore, I recommend the authors revise the manuscript for reconsideration. I have listed my comments below.

1. The Method section for air-liquid interface (ALI) culture does not describe the negative control sample (unstimulated) nor the number of replicate measurements. Please add detail on this. Was there one observation for each interferon molecule? Did you use the same negative control data for all differential expression analyses?

2. Line 119 says "p < 0.5" as a significance cutoff, which conflicts with Line 145 ("p < 0.05"). Please correct the manuscript if this is a typo.

3. It is important and common practice to apply multiple testing correction, especially for whole-genome scale measurements (a large number of hypothesis testing). There was no mention of multiple testing correction. What is the rationale for not using correction? If you do use correction, how many significantly differentially expressed genes survive the new cutoff criteria?

4. The heatmap plots (Fig. 1C, 2B, 2C, 2D, etc.) have multiple rows with the same gene name (for example, Fig 2C has multiple rows for FOXP2, TLE4, ZBTB20, etc.). Please explain this and revise the figures and text to clarify.

5. Also, all the heatmap plots use low-to-high abundance scale. Add description of how you calculate this relative scale in the Method section.

6. The section on string network analysis refers to the authors' previous publication, which is fine. Please add a brief description of analysis method in this manuscript for the readers such as input data (list of differentially expressed genes?), how STRING database is built, and why it is relevant.

7. One huge advantage of using a model system is that you can do lots of measurements that are not feasible with limited samples such as clinical samples. Have you performed any other experiments (microscopic imaging, TEER, etc) to corroborate your interpretation of gene expression data, especially on epithelial barrier integrity? 

8. The Introduction discusses airway epithelium structure. It will be very helpful for the readers if a schematic diagram showing epithelial structure/anatomy is included in this section.

9. Line 111: Please add the unit for 700 TEER measurements.

10. There are many typos throughout the manuscript. Please review your manuscript more thoroughly to fix them. Here are a few examples I found:

- Line 16: to examined -> to examine

- Line 34: TLR sense -> TLRs sense

- Line 68: adherend -> adherens 

- Line 92 and other occurrences: air-liquid interphase -> air-liquid interface 

- Line 94: differential -> differentially

- Line 97: positive -> positively 

- Line 181: Fig. 2C -> Fig. 2D 

- Line 323: house dust mice induced -> house dust mite-induced 

Author Response

Point-by-point Response

Date: 10/19/2022

Manuscript Number: biosensors-1945460

Title of Article: Genome-wide gene expression analysis reveals unique genes signatures of epithelial reorganization in primary airway epithelium induced by type-I, -II and -III Interferons

Name of Corresponding Author: Constanze A. Jakwerth, PhD

Email Address of Corresponding Author: constanze.jakwerth@tum.de

Reviewer #2:

The authors report differential gene expression and protein-protein interaction analyses of microarray data obtained from an airway epithelial model treated with different interferon molecules. While this reviewer agrees that understanding the impact of interferons in terms of antiviral activity and epithelial reorganization is important and useful to our readership, I think the manuscript is missing a few pieces of key information. Therefore, I recommend the authors revise the manuscript for reconsideration. I have listed my comments below.

We thank the reviewer for the overall positive feedback. Please find our response to each question below with line numbers corresponding to the lines in the clean version.

  1. The Method section for air-liquid interface (ALI) culture does not describe the negative control sample (unstimulated) nor the number of replicate measurements. Please add detail on this. Was there one observation for each interferon molecule? Did you use the same negative control data for all differential expression analyses?

We thank the reviewer for the constructive remark and have adapted the Methods section accordingly (Line 101-114): “Low passage primary human bronchial epithelial cells (NHBE, Lonza) from six genet-ically independent donors (n=6) were grown in PneumaCult-Ex Plus expansion medium (Stemcell) on corning transwell polyester membrane cell culture inserts (pre-coated with 1% collagen, Merck) according to the manufacturer´s instruction. Medium was applied to the basal and apical chamber until cells were grown 100% confluence. An airlift was performed by removing the apical medium and the basal medium was exchanged to Pneumacult-ALI maintenance medium (Stemcell). When the transepi-thelial electrical resistance (TEER), measured using EVOM2 instrument (World Preci-sion Instruments), reached the threshold of 700 Ω, cells each donor were stimulated with IFNα (300 IU/ml, Roche), IFNβ (100 IU/ml, Peprotech), IFNγ (200 IU/ml, Pro-mocell), IFNλ1 (100 ng/ml, Biotechne), IFNλ3 (100 ng/ml, Biotechne) or medium as control for 24h. Each negative control was genetically matched. TEER measurement was used to identify epithelial integrity prior stimulation and samples showing a TEER >700 Ω were classified as integer. Cells were harvested and RNA was extracted using AllPrep DNA/RNA Micro Kit (Qiagen).”

  1. Line 119 says "p < 0.5" as a significance cutoff, which conflicts with Line 145 ("p < 0.05"). Please correct the manuscript if this is a typo.

We thank the reviewer for identifying this typo, which we have now corrected (Line 128).

  1. It is important and common practice to apply multiple testing correction, especially for whole-genome scale measurements (a large number of hypothesis testing). There was no mention of multiple testing correction. What is the rationale for not using correction? If you do use correction, how many significantly differentially expressed genes survive the new cutoff criteria?

This problem of multiple testing has been widely discussed in the statistical literature: A correction for multiple testing generates additional problems: the universal null hypothesis is of little interest, the exact number of tests to be adjusted for cannot be determined, and the probability of type II error (false negative) increases. For these reasons, multiple testing correction is not recommended for genome-wide analysis tools (Gyorffy B, Gyorffy A, Tulassay Z. A "multiple testing" problémája és a genomiális kísérletekre alkalmazott megoldások [The problem of multiple testing and solutions for genome-wide studies]. Orv Hetil. 2005 Mar 20;146(12):559-63. Hungarian. PMID: 15853065.).

The decision on whether or not to apply a multiple testing correction must therefore depend on the biological material used and the scientific question. Two aspects are relevant for the current manuscript: 1) NHBEs show overall weak gene expression differences and high variability due to their primary nature. 2) In our case it is important to identify the quality of the effect of the different types of IFNs, and therefore our aim was for the reader to gain an overview picture of the biological mechanisms induced. Since multiple testing correction, such as the Benjamini Hochberg procedure or similar, eliminates a significant amount of expression differences, including genes that are well-described downstream target genes of the IFN signaling, such as ISGs (see Fig.1C), we decided to show all biologically relevant differentially expressed genes without multiple testing correction.

We have introduced a sentence in the Methods section to clarify this aspect for the reader (Line 129-131): “Due to low expression amplitude and variability of gene expression in primary cells, we did not apply multiple testing correction to the gene expression analysis.”

An alternative could be to display genes with >1.5-fold change but without statistical testing, eliminating the need for multiple testing correction. The disadvantage is that this results in very large figures, in which not all genes can be displayed and only selected ones would be shown and statistically tested. The message would be the same, but we will lose easy access to the data, as the figures would become more complex. We hope this has answered the reviewer’s question and are happy to provide modified figures if requested.

  1. The heatmap plots (Fig. 1C, 2B, 2C, 2D, etc.) have multiple rows with the same gene name (for example, Fig 2C has multiple rows for FOXP2, TLE4, ZBTB20, etc.). Please explain this and revise the figures and text to clarify.

Duplicate gene names indicate the abundance of two or more transcripts of the same gene in the analysis. These are now marked with a cross in each figure and referred to in the figure legends (Line 180; 207; 251; 280; 604).

  1. Also, all the heatmap plots use low-to-high abundance scale. Add description of how you calculate this relative scale in the Method section.

We appreciate this suggestion and have now added an explanation in the Method section (Line 143-145): “A 1.5-fold change above the “Normalized intensity values” in relation to the medium control were classified as high (red) abundance. This cut-off has been proven to be biologically meaningful in case airway epithelial cells.”

  1. The section on string network analysis refers to the authors' previous publication, which is fine. Please add a brief description of analysis method in this manuscript for the readers such as input data (list of differentially expressed genes?), how STRING database is built, and why it is relevant.

We thank the reviewer for this suggestion and introduced a sentence the method section (Line 150-152): “The STRING database uses open-source information about protein-protein interactions and connect this information with computational predictions to reveal physical and functional protein interactions.”

  1. One huge advantage of using a model system is that you can do lots of measurements that are not feasible with limited samples such as clinical samples. Have you performed any other experiments (microscopic imaging, TEER, etc) to corroborate your interpretation of gene expression data, especially on epithelial barrier integrity? 

Since these sophisticated 3D-ALI cultures were grown from rare primary material with laborious effort, including donor-dependent dropouts, we only included TEER measurements as an inclusion criterion. In future studies, we plan to expand functional measurements along with protein analyses to identify epithelial functions and viral defense mechanisms. We included the following sentence (Line 112-113): “TEER measurement was used to identify epithelial integrity prior stimulation and samples showing a TEER >700 Ω were classified as integer.”

  1. The Introduction discusses airway epithelium structure. It will be very helpful for the readers if a schematic diagram showing epithelial structure/anatomy is included in this section.

We thank the reviewer for this comment and have introduced a new Figure 1 into the Introduction section.

  1. Line 111: Please add the unit for 700 TEER measurements.

We thank the reviewer for this catch and have added the unit Ω to the Methods section in Line 109.

  1. There are many typos throughout the manuscript. Please review your manuscript more thoroughly to fix them. Here are a few examples I found:

We thank the reviewer for identifying these typos which we have now corrected.

- Line 16: to examined -> to examine : now Line 16

- Line 34: TLR sense -> TLRs sense : now Line 34

- Line 68: adherend -> adherens : now Line 63

- Line 92 and other occurrences: air-liquid interphase -> air-liquid interface : now Line 16; 84; 156; 171; 314; 327; 406

- Line 94: differential -> differentially : now Line 87

- Line 97: positive -> positively : now Line 90

- Line 181: Fig. 2C -> Fig. 2D : now Line 197

- Line 323: house dust mice induced -> house dust mite-induced : now Line 350

Round 2

Reviewer 2 Report

This reviewer finds the authors' reply addressed all of my comments and suggestions satisfactorily. Also, I appreciate that the authors included new supplemental tables of differentially expressed genes for all figures. I am certain the readers will benefit from the additional data and clarification.

I recommend this manuscript to be published in the current form (after copyediting of a few instances of misplaced hyphens).